# A sequential multi-target Mps1 phosphorylation cascade promotes spindle checkpoint signaling

Zhejian Ji[†], Haishan Gao[†], Luying Jia, Bing Li, Hongtao Yu*

Department of Pharmacology, Howard Hughes Medical Institute, University of Texas Southwestern Medical Center, Dallas, United States

**Abstract** The master spindle checkpoint kinase Mps1 senses kinetochore-microtubule attachment and promotes checkpoint signaling to ensure accurate chromosome segregation. The kinetochore scaffold Knl1, when phosphorylated by Mps1, recruits checkpoint complexes Bub1–Bub3 and BubR1–Bub3 to unattached kinetochores. Active checkpoint signaling ultimately enhances the assembly of the mitotic checkpoint complex (MCC) consisting of BubR1–Bub3, Mad2, and Cdc20, which inhibits the anaphase-promoting complex or cyclosome bound to Cdc20 (APC/C$^{Cdc20}$) to delay anaphase onset. Using in vitro reconstitution, we show that Mps1 promotes APC/C inhibition by MCC components through phosphorylating Bub1 and Mad1. Phosphorylated Bub1 binds to Mad1–Mad2. Phosphorylated Mad1 directly interacts with Cdc20. Mutations of Mps1 phosphorylation sites in Bub1 or Mad1 abrogate the spindle checkpoint in human cells. Therefore, Mps1 promotes checkpoint activation through sequentially phosphorylating Knl1, Bub1, and Mad1. This sequential multi-target phosphorylation cascade makes the checkpoint highly responsive to Mps1 and to kinetochore-microtubule attachment.

*For correspondence: hongtao. yu@utsouthwestern.edu

[†]These authors contributed equally to this work

Competing interests: The authors declare that no competing interests exist.

## Introduction

Faithful chromosome segregation is ensured by a cellular surveillance system termed the spindle checkpoint (*Foley and Kapoor, 2013*; *Jia et al., 2013*; *Musacchio, 2015*; *Sacristan and Kops, 2015*). During mitosis, the spindle checkpoint senses kinetochores that are not attached or improperly attached to spindle microtubules and enhances the production of the mitotic checkpoint complex (MCC) consisting of BubR1–Bub3, Mad2, and Cdc20 (*Fang, 2002*; *Sudakin et al., 2001*; *Tang et al., 2001*). MCC inhibits the ubiquitin ligase activity of the anaphase-promoting complex or cyclosome bound to its activator Cdc20 (APC/C$^{Cdc20}$) (*Alfieri et al., 2016*; *Izawa and Pines, 2015*; *Yamaguchi et al., 2016*). Inhibition of APC/C$^{Cdc20}$ delays anaphase onset until all kinetochores in a mitotic cell achieve proper kinetochore-microtubule attachment. A dysfunctional spindle checkpoint causes chromosome missegregation and aneuploidy, which have been implicated in cancers, birth defects, and other human diseases (*Gorbsky, 2015*; *Holland and Cleveland, 2012*).

The kinetochore-microtubule attachment is closely monitored by the evolutionarily conserved checkpoint kinase Mps1 (*Aravamudhan et al., 2015*; *Hiruma et al., 2015*; *Ji et al., 2015*; *Liu and Winey, 2012*). Mps1 recognizes unattached kinetochores through binding to the critical microtubule receptor, the Ndc80 complex (Ndc80C), when it is not bound by microtubules (*Hiruma et al., 2015*; *Ji et al., 2015*). Ndc80C-bound Mps1 initiates checkpoint signaling by phosphorylating multiple methionine-glutamate-leucine-threonine (MELT) motifs of the kinetochore scaffold Knl1 (*London et al., 2012*; *Shepperd et al., 2012*; *Yamagishi et al., 2012*). Phosphorylated MELT motifs recruit checkpoint complexes Bub1–Bub3 and BubR1–Bub3 to the kinetochores (*Krenn et al., 2014*; *Overlack et al., 2015*; *Primorac et al., 2013*; *Vleugel et al., 2013*; *Zhang et al., 2014*). Bub1 and

BubR1 in turn recruit and modify Cdc20 (*Di Fiore et al., 2015*; *Diaz-Martinez et al., 2015*; *Han et al., 2014*; *Jia et al., 2016*; *Lischetti et al., 2014*). Mad1–Mad2 is also targeted to kinetochores through multiple mechanisms (*Caldas et al., 2015*; *Kim et al., 2012*; *London and Biggins, 2014*; *Matson and Stukenberg, 2014*; *Silió et al., 2015*), and promotes the conformational activation of open Mad2 (O-Mad2) to intermediate Mad2 (I-Mad2) (*De Antoni et al., 2005*; *Hara et al., 2015*; *Luo et al., 2002, 2004*; *Luo and Yu, 2008*; *Mapelli et al., 2007*; *Mapelli and Musacchio, 2007*; *Sironi et al., 2002*). I-Mad2 is passed on to Cdc20 to form the Cdc20–C-Mad2 complex, which subsequently associates with BubR1–Bub3 to form MCC (*Kulukian et al., 2009*). MCC diffuses from kinetochores to inhibit APC/C$^{Cdc20}$ throughout the cell.

Artificial tethering of Mad1–Mad2 to attached kinetochores causes prolonged activation of the spindle checkpoint and delays anaphase onset in the absence of spindle poisons (*Maldonado and Kapoor, 2011*). This artificial activation of the checkpoint still requires the kinase activity of Mps1, indicating that, aside from Knl1 phosphorylation, Mps1 has downstream functions in the checkpoint. Studies in the budding and fission yeast suggest that one such function of Mps1 is to phosphorylate Bub1 and establish a phosphorylation-dependent interaction between Bub1 and Mad1 (*London and Biggins, 2014*; *Mora-Santos et al., 2016*). Although the RLK motif in the Mad1 C-terminal domain (CTD) responsible for phospho-Bub1 binding is conserved from yeast to man (*Brady and Hardwick, 2000*; *Kim et al., 2012*), this Mps1-stimulated Bub1–Mad1 interaction has not been formally demonstrated in humans. Whether Mps1 has additional roles in regulating Mad1 also remains to be established.

Here, we report a direct phosphorylation-dependent interaction between human Bub1 and Mad1 in vitro. This interaction requires phosphorylation of Bub1 by both Cdk1 and Mps1, and is critical for spindle checkpoint activation in human cells. We have in vitro reconstituted checkpoint-dependent APC/C$^{Cdc20}$ inhibition by MCC components, which requires a functional phospho-Bub1–Mad1 scaffold. Furthermore, we have found that Mps1 directly phosphorylates Mad1 in its CTD. Phosphorylated Mad1 CTD directly binds to the N-terminal region of Cdc20 and contributes to APC/C$^{Cdc20}$ inhibition presumably through stimulating MCC assembly. Therefore, Mps1 activates the spindle checkpoint through a three-step phosphorylation cascade. Phosphorylation of Knl1 by Mps1 enables Bub1 recruitment. Phosphorylation of Bub1 by Mps1 establishes the Bub1–Mad1 interaction. Phosphorylation of Mad1 by Mps1 positions Cdc20 for MCC assembly. We propose that this sequential multi-target phosphorylation cascade makes the checkpoint highly responsive to Mps1 whose function is directly regulated by kinetochore-microtubule attachment.

## Results

### Phosphorylation of Bub1 by Cdk1 is required for the spindle checkpoint

Bub1 is a spindle checkpoint kinase conserved from yeast to man. Aside from its C-terminal kinase domain, the N-terminal non-kinase region of Bub1 contains multiple functional domains or motifs, including the tetratricopeptide repeat (TPR) domain for Knl1 binding, the Gle2-binding sequence (GLEBS) for Bub3 binding, the conserved middle region (CM; residues 458–476), and the Cdc20-binding phenylalanine box (Phe box; also known as ABBA motif) and lysine-glutamate-asparagine (KEN) box (*Di Fiore et al., 2015*; *Diaz-Martinez et al., 2015*; *Kang et al., 2008*) (*Figure 1A*). Bub1 is highly phosphorylated in mitosis (*Chen, 2004*; *Tang et al., 2004*). We immunoprecipitated the endogenous Bub1 protein from mitotic HeLa cell lysates, and analyzed its mitotic phosphorylation sites with mass spectrometry. We identified 30 phosphorylated residues that were scattered in the non-kinase domain of Bub1 (*Figure 1—figure supplement 1A*). One such site, S459, is located in the Bub1 CM.

We generated a phospho-specific antibody against Bub1 phospho-S459 (pS459). Recombinant Bub1$^{\Delta K}$–Bub3 was strongly phosphorylated at S459 by recombinant Cdk1, but not Mps1, in vitro (*Figure 1B*). This finding is consistent with the fact that Bub1 S459 is followed by a proline, and matches the minimal Cdk1 substrate consensus of [S/T]P. We next confirmed that Myc-Bub1 was indeed phosphorylated in mitotic HeLa cells (*Figure 1C*). Interestingly, Bub1 S459 phosphorylation in mitosis was reduced upon treatment of the Mps1 inhibitor, reversine. Because recombinant Bub1 protein was not phosphorylated at S459 by Mps1 in vitro, the effect of Mps1 inhibition on S459 phosphorylation may be indirectly through checkpoint inactivation. This result further suggests that

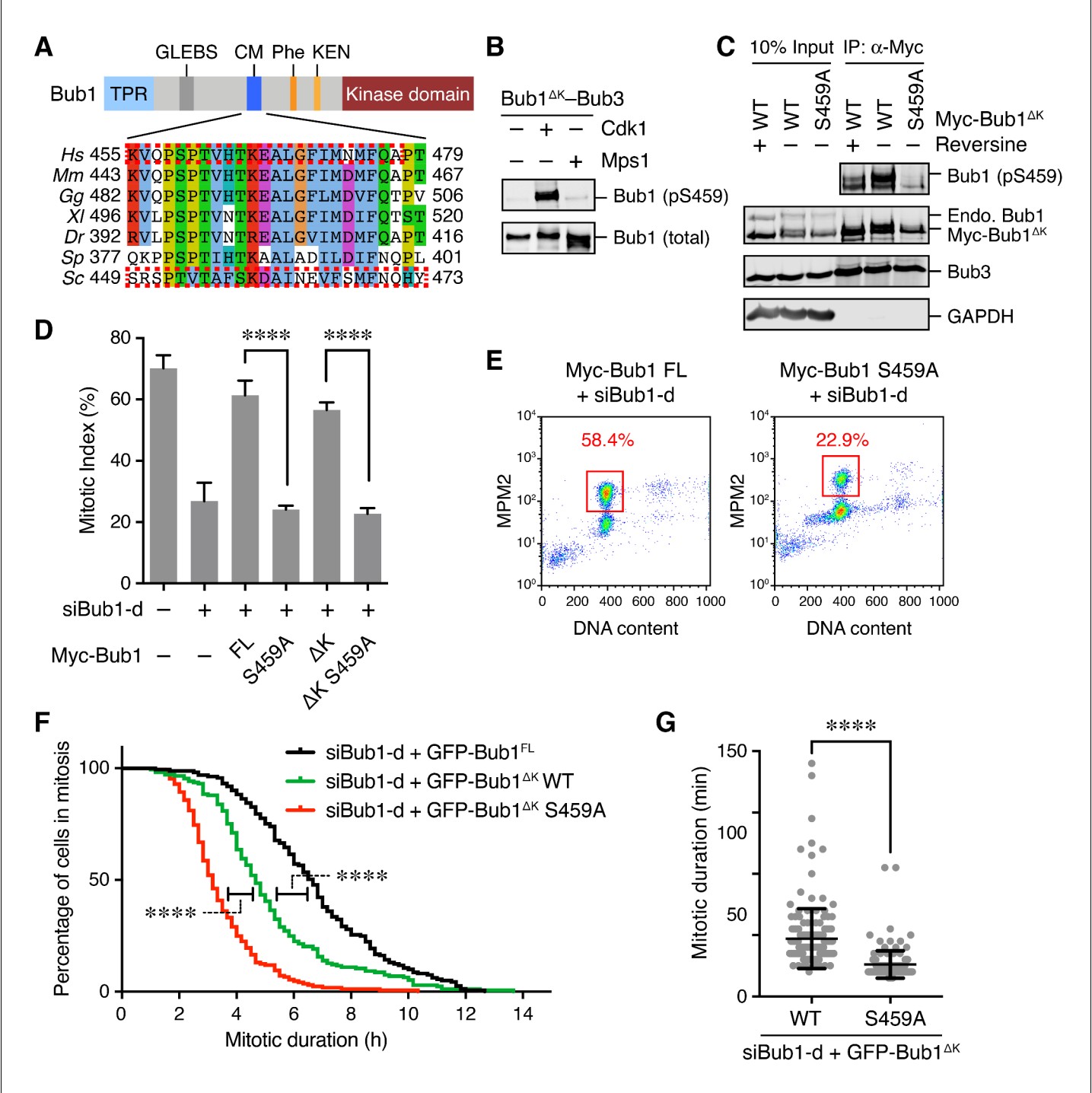

**Figure 1.** Phosphorylation of Bub1 S459 is critical for spindle checkpoint activation. (A) Domains and motifs of Bub1 and sequence alignment of its conserved motif (CM). TPR, tetratricopeptide repeat; GLEBS, Gle2-binding sequence; Phe, phenylalanine-containing motif, also known as 'ABBA' motif; KEN, lysine-glutamate-asparagine motif. *Hs, Homo sapiens; Mm, Mus musculus; Gg, Gallus gallus; Xl, Xenopus laevis; Dr, Danio rerio; Sp, Schizosaccharomyces pombe; Sc, Saccharomyces cerevisiae.* The boxed regions in scBub1 and hsBub1 were synthesized as phospho-peptides and used in this study. (B) Recombinant Bub1$^{\Delta K}$–Bub3 complex was incubated with Cdk1–Cyclin B1 (Cdk1) or Mps1 in the presence of ATP. The kinase reactions were blotted with indicated antibodies. (C) HeLa cells stably expressing Myc-Bub1$^{\Delta K}$ wild-type (WT) or S459A mutant were treated with nocodazole and MG132 in the presence or absence of reversine. Myc-Bub1 was immunoprecipitated and blotted with indicated antibodies. Endo., endogenous. (D) Flow cytometry of HeLa Tet-On cells stably expressing indicated siRNA-resistant Myc-Bub1 transgenes that were transfected with siBub1-d and treated with taxol. Mitotic indices were calculated as percentages of MPM2$^{+}$ 4N cells in flow cytometry, and then plotted. FL, full-length.

*Figure 1 continued on next page*

*Figure 1 continued*

△K, mutant with the kinase domain truncated. Error bars, s.d. (n = 4 independent experiments). ****p<0.0001; Student's t-test. (E) Representative flow cytometry plots of cells in (D). (F) HeLa Tet-On cells stably expressing indicated GFP-Bub1 transgenes were transfected with siBub1-d, treated with taxol, and imaged with time lapse microscopy. Cumulative percentages of cells remaining in mitosis were plotted against mitotic duration. Data from three independent experiments were combined. n (FL) = 161; n (△K)=173; n (△K S459A)=169. ****p<0.0001; Log-rank test. (G) Mitotic durations of cells stably expressing GFP-Bub1$^{\triangle K}$ WT or S459A that were depleted of endogenous Bub1 and not treated with microtubule poisons. Data from three independent experiments were combined. Each dot represents one cell. n (WT) = 170; n (S459A)=157. ****p<0.0001; Student's t-test.

The following figure supplement is available for figure 1:

**Figure supplement 1.** Identification of mitotic phosphorylation sites in human Bub1 and characterization of the Bub1 S459A mutant.

Bub1 S459 phosphorylation is regulated by the spindle checkpoint, possibly through the actions of phosphatases.

We then tested whether this phosphorylation site was required for the spindle checkpoint. HeLa cells depleted of endogenous Bub1 with a small interfering RNA (siRNA) had reduced mitotic index in the presence of taxol, indicative of checkpoint defects (*Figure 1D*). This mitotic arrest deficiency was rescued by ectopic expression of siRNA-resistant Bub1 full-length (FL), but not the S459A mutant (*Figure 1D and E*, and *Figure 1—figure supplement 1B*), suggesting a requirement for Bub1 S459 phosphorylation in the spindle checkpoint.

In mitosis, Bub1 phosphorylates histone H2A and Cdc20 (*Kawashima et al., 2010*; *Tang et al., 2004*). While H2A phosphorylation by Bub1 contributes to Shugoshin recruitment to kinetochores and protection of centromeric cohesion (*Kawashima et al., 2010*; *Liu et al., 2013a*, *2015*, *2013b*), Bub1-mediated phosphorylation of Cdc20 contributes to the spindle checkpoint (*Jia et al., 2016*; *Kang et al., 2008*; *Tang et al., 2004*). We thus tested whether Bub1 S459 phosphorylation regulated its kinase activity. Bub1 FL and S459A exhibited comparable kinase activities towards Cdc20 S153 in vitro (*Figure 1—figure supplement 1C*). Therefore, Bub1 S459 phosphorylation might contribute to checkpoint activation without compromising its kinase activity.

Bub1 is known to have a scaffolding role in the spindle checkpoint (*Jia et al., 2016*; *Klebig et al., 2009*). Indeed, expression of the Bub1 mutant lacking the kinase domain (Bub1$^{\triangle K}$) at higher levels restored the taxol-induced mitotic arrest of Bub1-depleted cells (*Figure 1D* and *Figure 1—figure supplement 1B*). Bub1$^{\triangle K}$ S459A failed to rescue the mitotic arrest deficiency caused by Bub1 depletion. This result suggests that the scaffolding function of the non-kinase region of Bub1 requires S459 phosphorylation.

To further confirm a role of Bub1 S459 phosphorylation in checkpoint-dependent mitotic arrest, we performed live-cell imaging of HeLa cells stably expressing GFP Bub1$^{FL}$, Bub1$^{\triangle K}$, or Bub1$^{\triangle K}$ S459A that were depleted of endogenous Bub1. In the presence of taxol, cells expressing Bub1$^{FL}$ or Bub1$^{\triangle K}$ stayed in mitosis for about 6.7 hr or 4.7 hr, respectively, before undergoing mitotic cell death or slippage. Cells expressing Bub1$^{\triangle K}$ S459A, however, escaped from mitosis after only about 3.2 hr (*Figure 1F* and *Figure 1–figure supplement 1D*). Furthermore, in an unperturbed cell cycle, cells expressing Bub1$^{\triangle K}$ S459A had a shorter mitotic duration (26 min), as compared to cells expressing Bub1$^{\triangle K}$ (47 min) (*Figure 1G*). Collectively, these lines of evidence suggest the non-kinase region of Bub1 has an important role in supporting the spindle checkpoint, and this role requires phosphorylation of S459 by Cdk1.

## Phosphorylation of yeast Bub1 by Mps1 enables its binding to Mad1

It has been reported previously that the middle region of *Saccharomyces cerevisiae* Bub1 (scBub1), when phosphorylated by Mps1, binds to Mad1–Mad2 and that this phosphorylation-dependent Bub1–Mad1 interaction is critical for the spindle checkpoint (*London and Biggins, 2014*). An attractive hypothesis is that phosphorylation of S459 in human Bub1 by Cdk1 similarly enables its binding to Mad1. However, despite repeated attempts, we could not detect binding between Mad1 and recombinant Bub1$^{\triangle K}$–Bub3 phosphorylated by Cdk1 in vitro. We decided to first examine whether phosphorylation of T453 of scBub1 (which corresponds to S459 in human Bub1) was required for binding to scMad1. Because the full-length Mad1 was difficult to express, we used a C-terminal fragment of Mad1 (Mad1$^E$) that contains both the Mad2-interacting motif (MIM) and the C-terminal

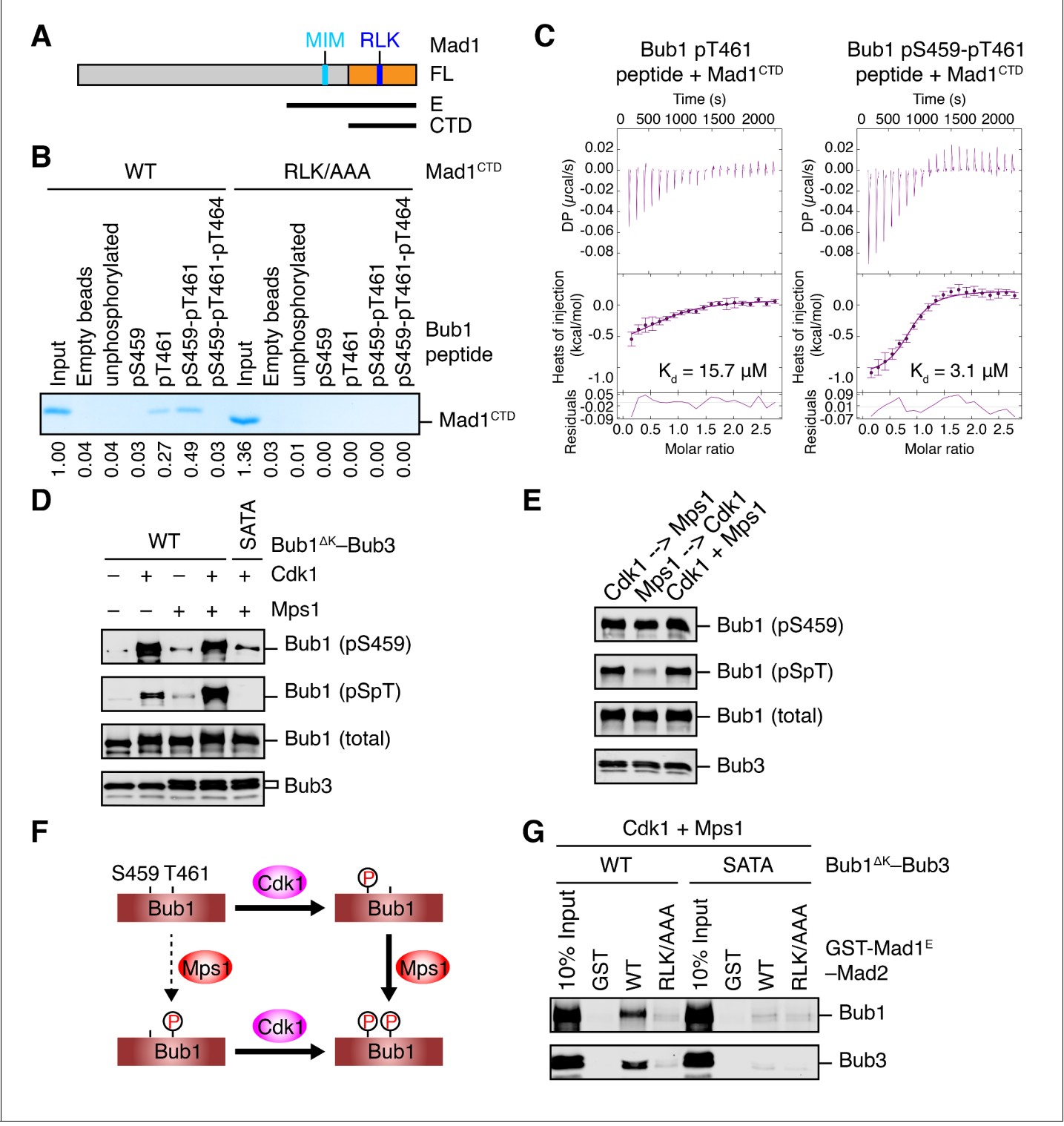

**Figure 2.** Sequential phosphorylation of human Bub1 by Cdk1 and Mps1 enhances its binding to Mad1. (**A**) Domains and motifs of Mad1. Schematic domain structures and tested fragments of Mad1 protein. CTD, C-terminal domain; MIM, Mad2-interacting motif; RLK, the arginine-leucine-lysine motif. The E fragment of Mad1 containing both MIM and CTD was used to make the Mad1–Mad2 complex in this study. (**B**) In vitro pull-down of Mad1$^{CTD}$ using empty beads or beads conjugated to the indicated Bub1 peptides. Proteins bound on beads were separated on SDS-PAGE and visualized by Coomassie blue staining. Relative band intensities were quantified and indicated below the gel. (**C**) Isothermal titration calorimetry (ITC) assays of binding between the C-terminal domain (CTD) of human Mad1 and the human Bub1 peptides containing either phospho-T461 alone or both phospho-S459 and phospho-T461. $K_d$, dissociation constant. (**D**) In vitro kinase assays of recombinant Bub1$^{\Delta K}$–Bub3 WT or S459A/T461A (SATA) treated with

*Figure 2 continued on next page*

*Figure 2 continued*

Cdk1 or Mps1 or both. The kinase reactions were resolved on SDS-PAGE and blotted with indicated antibodies. pSpT, a phospho-specific antibody recognizing both phospho-S459 and phospho-T461. (E) In vitro kinase assays similar to (D), except that the kinases were added in the indicated orders. In lanes 1 and 2, Cdk1 or Mps1 was first incubated with the substrate for 30 min before being inhibited by RO3306 or reversine, respectively. (F) Schematic drawing of the sequential phosphorylation of Bub1 at S459 and T461 by Cdk1 and Mps1. (G) Bub1$^{\Delta K}$–Bub3 WT and SATA were first phosphorylated by both Cdk1 and Mps1, and then assayed for binding to GST-Mad1$^E$–Mad2 beads. The bound proteins were blotted with the indicated antibodies.

The following figure supplement is available for figure 2:

**Figure supplement 1.** Binding of scBub1 phosphorylated at T455 by Mps1 to scMad1 CTD.

domain (CTD) (*Figure 2A*). Co-expression of scBub1$_{369-608}$ with the kinase domain of scMps1 in bacteria led to phosphorylation of the scBub1 fragment (*Figure 2—figure supplement 1A*). Consistent with a previous report (*London and Biggins, 2014*), only the phosphorylated scBub1 bound to the scMad1$^E$–scMad2 complex. Internal deletion of the CM rendered scBub1$_{369-608}$ unable to bind to scMad1–Mad2, indicating a requirement of this motif in this interaction. Isothermal titration calorimetry (ITC) analysis showed that the smaller phosphorylated scBub1$_{449-530}$ fragment containing the CM bound to scMad1–Mad2 with a moderate affinity of 2.2 μM (*Figure 2—figure supplement 1B*). Interestingly, this phospho-scBub1 fragment bound to the CTD of scMad1 with a similar affinity, indicating that Mad1 CTD is sufficient for the scBub1–scMad1 interaction, and Mad2 and the MIM of Mad1 are not required for this interaction.

There are three conserved serine/threonine residues in the CM of scBub1: T453, T455 and S458. Our mass spectrometry analysis revealed that all three residues were phosphorylated in scBub1$_{449-530}$ co-expressed with scMps1. We next synthesized scBub1 CM phospho-peptides containing pT453 or pT455 or both. We could not test the contribution of scBub1 pS458 to scMad1 binding, because the synthesis of pS458-containing peptides failed for unknown reasons. Both the singly phosphorylated pT455 peptide and the doubly phosphorylated pT453-pT455 peptide, but not the singly phosphorylated pT453 peptide, bound efficiently to scMad1 CTD (*Figure 2—figure supplement 1C*). Mutating the conserved RLK motif in Mad1 CTD abolished scMad1 binding to both scBub1 phospho-CM peptides. Furthermore, ITC analysis showed that the pT455 and pT453-pT455 peptides bound to scMad1 CTD with the same affinity (*Figure 2—figure supplement 1D*). Thus, Mps1-dependent phosphorylation of scBub1 at T455 alone is sufficient to establish scBub1–scMad1 binding. The scBub1–scMad1 interaction is mainly mediated by phospho-T455 of scBub1 and the RLK motif of scMad1. We note that phosphorylation of the T455 site in scBub1 has been previously reported (*London and Biggins, 2014*), and mutation of this residue disrupts scBub1–scMad1 binding and causes benomyl sensitivity in yeast cells, establishing the functional relevance of this phosphorylation event.

## Sequential phosphorylation of human Bub1 by Cdk1 and Mps1 enhances Mad1 binding

Because phospho-T455 of scBub1 (which corresponds to T461 in human Bub1) was more critical for scMad1 binding than phospho-T453 (which corresponds to S459 in human Bub1), we wondered if phosphorylation of T461 in human Bub1 was similarly required for human Mad1 binding. We synthesized several phospho-peptides of human Bub1 CM and tested their binding to human Mad1. The pT461 peptide exhibited weak binding to human Mad1 CTD, and the doubly phosphorylated pS459-pT461 peptide had stronger binding (*Figure 2B*). ITC assays confirmed that human Mad1 bound with higher affinity to the pS459-pT461 peptide than to the pT461 peptide (*Figure 2C*). As a specificity control, binding of both peptides was reduced by mutating the RLK motif in Mad1 CTD. Interestingly, the triply phosphorylated pS459-pT461-pT464 peptide bound more weakly to Mad1 than the pS459-pT461 peptide, indicating that phosphorylation of T464 had an inhibitory effect. Our results suggest that phosphorylation of both S459 and T461 of human Bub1 is required for its optimal interaction with Mad1.

We generated a phospho-specific antibody that recognized both pS459 and pT461 (pSpT) in Bub1. Incubation of recombinant Bub1$^{\Delta K}$–Bub3 with both Cdk1 and Mps1 led to efficient phosphorylation of both residues, whereas incubation with either kinase did not (*Figure 2D*). The pSpT

antibody did not recognize the Bub1$^{\Delta K}$S459A/T461A (SATA) mutant treated with both Cdk1 and Mps1, demonstrating the specificity of the antibody. The weak signal seen with Cdk1 alone was likely due to cross-reactivity of the pSpT antibody with Bub1 singly phosphorylated at S459. We then tested the order of these two Cdk1- and Mps1-mediated phosphorylation events. When we first incubated Bub1$^{\Delta K}$ with Cdk1 to phosphorylate S459 and then inhibited Cdk1 with RO3306, subsequent addition of Mps1 led to efficient phosphorylation at T461, as detected by the pSpT antibody (*Figure 2E*). In contrast, when we first incubated Bub1$^{\Delta K}$ with Mps1, inhibited Mps1 with reversine, and then added Cdk1, Bub1$^{\Delta K}$ was still efficiently phosphorylated at S459, but was only weakly phosphorylated at T461, as revealed by a weak pSpT signal. This result suggests that phosphorylation of Bub1 at S459 and T461 by Cdk1 and Mps1 occurs sequentially (*Figure 2F*). Cdk1 phosphorylation at S459 primes Mps1 phosphorylation at T461. This notion is consistent with the fact that Mps1 prefers to phosphorylate sites with an acidic residue at the −2 position (*Dou et al., 2011*).

With both Cdk1- and Mps1-mediated phosphorylation, we observed an interaction between human Bub1$^{\Delta K}$–Bub3 and GST-Mad1$^E$–Mad2 in vitro (*Figure 2G*). As important controls, mutations of both phosphorylation sites on Bub1 or the RLK motif on Mad1 greatly reduced this interaction. Thus, the phosphorylation-dependent interaction between Bub1 CM and Mad1 CTD is conserved from yeast to man.

## Bub1–Mad1 binding is critical for spindle checkpoint activation in human cells

Although we did not identify T461 as a phosphorylation site on endogenous human Bub1 isolated from mitotic HeLa cells, Bub1 T461 had been identified as a phospho-residue in a phospho-proteomics study (*Daub et al., 2008*). Using the pSpT phospho-specific antibody, we confirmed that both S459 and T461 sites were phosphorylated in checkpoint-active cells (*Figure 3A*). Interestingly, the T461A mutation also greatly reduced the pS459 signal. We do not know whether this loss of pS459 signal was due to the dephosphorylation of this site or a direct effect of this mutation on antibody recognition.

We next tested whether phosphorylation of Bub1 T461 was required for the spindle checkpoint in human cells. Similar to Bub1 S459A, neither Bub1 T461A nor Bub1 SATA could rescue the mitotic arrest deficiency caused by depletion of endogenous Bub1 (*Figure 3B and C*). Moreover, as we showed previously (*Jia et al., 2016*), partial depletion of Bub1 and inhibition of Plk1 by BI 2536 synergistically inactivated the spindle checkpoint and reduced nocodazole-triggered mitotic arrest (*Figure 3B and C*). This mitotic arrest deficiency was rescued by the expression of siRNA-resistant Bub1 WT, but not S459A, T461A, or SATA, indicating that phosphorylation of both sites on Bub1 are critical for checkpoint signaling.

We further tested whether the RLK motif in Mad1 (which was likely the binding element of the Bub1 phospho-residues) was similarly required for the spindle checkpoint. Because we could not deplete Mad1 to sufficiently low levels to reveal a defect in nocodazole-dependent mitotic arrest, we adopted a previously reported strategy that involved the artificial targeting of Mad1 to kinetochores through a fusion to Mis12, a subunit of the outer kinetochore Mis12 complex (*Figure 3D*) (*Maldonado and Kapoor, 2011*). Expression of the Mis12–Mad1 fusion protein caused spontaneous mitotic arrest, which was dependent on Mps1, BubR1, and Mad2, indicating that this arrest required an active spindle checkpoint (*Figure 3D and E*). Consistent with previous studies in yeast and human cells (*Ballister et al., 2014*; *Heinrich et al., 2014*), the Mis12–Mad1 fusion proteins with the MIM or RLK motifs mutated failed to induce the mitotic arrest. Thus, both the Mad2-binding and Bub1-binding activities are critical for the checkpoint function of Mad1. Taken together, our results establish the functional importance of the phosphorylation-dependent Bub1–Mad1 interaction in the spindle checkpoint in human cells.

## Bub1–Mad1 acts as a scaffold to promote APC/C inhibition by MCC components

Next, we investigated the mechanism by which the Bub1–Mad1 interaction contributed to checkpoint activation. Although neither Bub1 nor Mad1 is a component of MCC, they both directly interact with MCC components. The constitutive Mad1–C-Mad2 complex interacts with O-Mad2 and

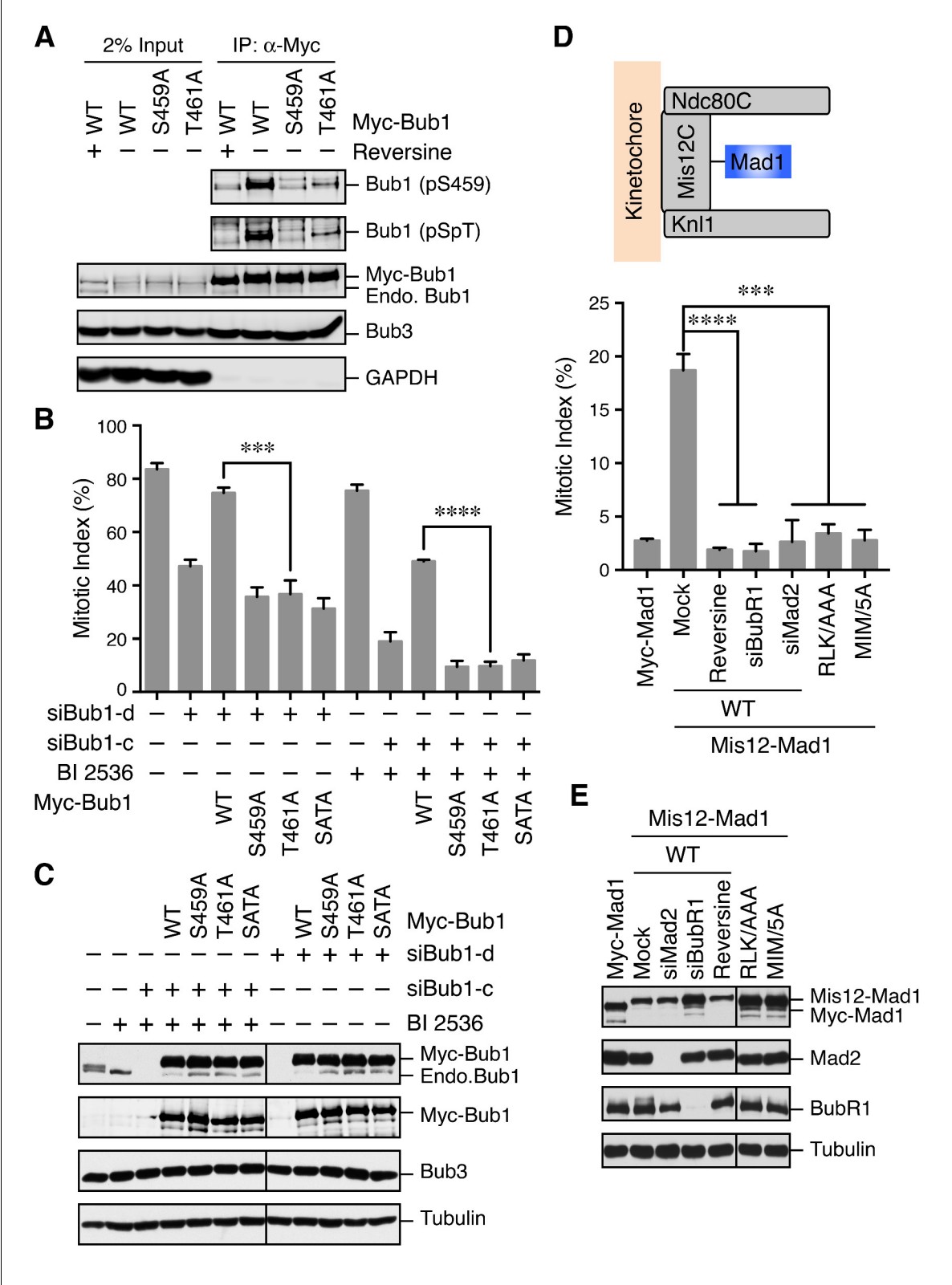

**Figure 3.** The Bub1–Mad1 interaction is crucial for checkpoint activation in human cells. (**A**) HeLa cells expressing Myc-Bub1 transgenes were treated with nocodazole and MG132 in the presence or absence of reversine. Myc-Bub1 proteins were immunoprecipitated and blotted with the indicated antibodies. (**B**) Mitotic indices of cells expressing Bub1 transgenes that were transfected with the indicated Bub1 siRNA and treated with nocodazole in the presence or absence of the Plk1 inhibitor BI2536. Error bars, s.d. (n = 3 independent experiments). ***p<0.001; ****p<0.0001; Student's t-test. (**C**)

*Figure 3 continued on next page*

Figure 3 continued

Lysates of cells in (B) were blotted with the indicated antibodies. Endo., endogenous. (D) Mitotic indices of cells expressing the indicated Mis12–Mad1 fusion proteins that were treated with reversine or siRNAs against BubR1 or Mad2. MIM/5A, Mad1 mutant with its MIM ($^{541}$KVLHM$^{545}$) changed to five alanines. Error bars, s.d. (n = 3 independent experiments). ***p<0.001; ****p<0.0001; Student's t-test. (E) Lysates of cells in (D) were blotted with the indicated antibodies.

converts it to I-Mad2, which can bind to Cdc20 to form the Cdc20–C-Mad2 complex. Furthermore, several previous studies have characterized the direct, checkpoint-relevant binding of Bub1 to Cdc20 (*Di Fiore et al., 2015*; *Diaz-Martinez et al., 2015*; *Kang et al., 2008*). These results raise the intriguing possibility that the Bub1–Mad1 complex functions as a scaffold to recruit MCC components (particularly I-Mad2 and Cdc20) into close proximity, thus promoting MCC assembly.

To test this hypothesis, we decided to reconstitute the process of MCC assembly and subsequent APC/C$^{Cdc20}$ inhibition using a pre-assembled phospho-Bub1–Mad1 complex (*Figure 4A*). For this purpose, recombinant full-length (FL), functional domains, or the relevant binary complexes of Bub1, Bub3, Mad1, Cdk1, Mps1, BubR1, Mad2, and Cdc20 were expressed in bacteria or insect cells and purified (*Figure 4B*). The N-terminal fragment of BubR1 (residues 1–370, BubR1$^N$) has been shown previously to support the formation of functional MCC (*Jia et al., 2016*; *Lara-Gonzalez et al., 2011*). We separated monomeric and dimeric forms of Mad2 with ion exchange chromatography (*Figure 4—figure supplement 1A*) (*Luo et al., 2004*). The monomeric Mad2 (Mad2$^{Mono}$) is mostly the inactive O-Mad2, while the dimeric Mad2 (Mad2$^{Di}$) contains a mixture of dimers of active C-Mad2 and asymmetric dimers between C-Mad2 and I-Mad2 (*Hara et al., 2015*; *Yang et al., 2008*).

We first compared the activities of Mad2$^{Mono}$ and Mad2$^{Di}$ to assemble functional MCC that was capable of inhibiting APC/C$^{Cdc20}$, in the absence of Bub1$^{ΔK}$–Bub3, Mad1$^E$–Mad2, Cdk1, and Mps1 (*Figure 4—figure supplement 1B*). Incubation of BubR1$^N$ and Cdc20$^{ΔN60}$ with Mad2$^{Di}$ allowed the spontaneous formation of functional MCC that inhibited securin ubiquitination by APC/C$^{Cdc20}$. Mad2$^{Mono}$ was much less efficient in forming functional MCC and only showed weak APC/C$^{Cdc20}$ inhibition at very high doses. This result suggests that the conformational change of Mad2 is a rate-limiting step in MCC assembly. Thus, the APC/C inhibition assay involving the use of Mad2$^{Mono}$ can be employed to test the conformational activation of Mad2 and subsequent MCC assembly in the presence of the pre-assembled Bub1–Mad1 complex.

Bub1$^{ΔK}$–Bub3 was first phosphorylated extensively by Cdk1 and Mps1 kinases and then mixed with the Mad1$^E$–Mad2 complex to form the Bub1–Mad1 complex. Subsequently, the mixture of Bub1$^{ΔK}$–Bub3, Mad1$^E$–Mad2, Cdk1, and Mps1 (hereafter referred to as the Bub1–Mad1 mixture) were mixed with BubR1$^N$, Cdc20$^{ΔN60}$, and Mad2$^{Mono}$ (referred to as the MCC mixture) to allow MCC formation (*Figure 4A*). (The reason for using Cdc20$^{ΔN60}$ as opposed to Cdc20$^{FL}$ will become apparent later.) The entire reaction mixture was then assayed for its ability to inhibit APC/C$^{Cdc20}$. Strikingly, securin ubiquitination by APC/C$^{Cdc20}$ was strongly inhibited when the MCC mixture containing Mad2$^{Mono}$ was pre-incubated with the WT Bub1–Mad1 mixture (*Figure 4C*). Omitting Mad2$^{Mono}$ abolished APC/C inhibition, indicating that MCC was formed by the activated Mad2$^{Mono}$, and not by the C-Mad2 bound to Mad1. This finding is consistent with the in vivo observation that Mad1 over-expression abrogates the spindle checkpoint, presumably by titrating free Mad2 (*Chung and Chen, 2002*). Remarkably, Mps1 inhibition by reversine, mutations of S459 and T461 in Bub1 (SATA), or mutation of the Mad1 RLK motif (RLK/AAA) all greatly weakened APC/C$^{Cdc20}$ inhibition by the MCC mixture. These results indicate that the intact phosphorylation-dependent Bub1–Mad1 interaction promotes APC/C$^{Cdc20}$ inhibition by the MCC mixture, presumably through facilitating Mad2 activation and MCC assembly.

## Phosphorylation of Mad1 by Mps1 facilitates MCC assembly and checkpoint activation

When testing different conditions for the in vitro reconstitution of Mps1-dependent APC/C inhibition by MCC, we noticed that, in the presence of only Mad1$^E$–Mad2 (without Bub1), Mps1 stimulated APC/C$^{Cdc20}$ inhibition by the MCC mixture containing Cdc20$^{FL}$ (*Figure 5A and B*). Interestingly, this Bub1-independent stimulatory effect of Mps1 was not observed with the MCC mixture containing

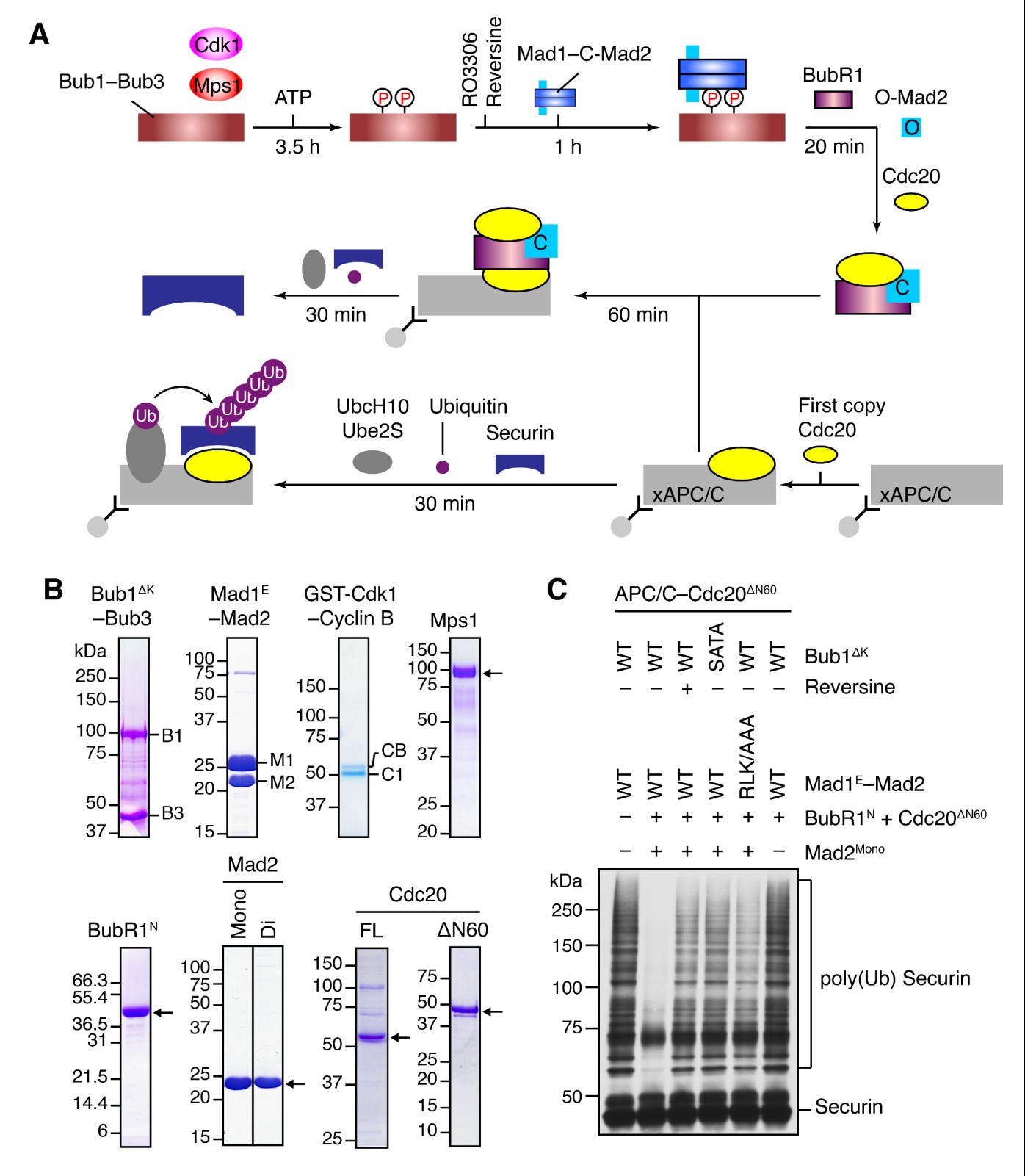

**Figure 4.** The Bub1–Mad1 complex promotes APC/C$^{Cdc20}$ inhibition by MCC components. (**A**) Flow charts of the in vitro reconstitution of Mps1-stimulated APC/C inhibition by MCC components. The incubation times of each reaction step are indicated. All processes were performed at room temperature. Molecules are not drawn to scale. xAPC/C, the APC/C complex isolated from *Xenopus* egg extract by immunoprecipitation. Ub, ubiquitin. (**B**) A collection of recombinant proteins used for the in vitro reconstitution. Relevant protein bands were labeled or indicated by arrows. B1, Bub1$^{\Delta K}$; *Figure 4 continued on next page*

*Figure 4 continued*

B3, Bub3; M1, Mad1$^E$; M2, Mad2; CB, Cyclin B1; C1, GST-Cdk1; Mono, monomeric Mad2; Di, dimeric Mad2; FL, full-length. (**C**) The ubiquitination reactions as depicted in (A) were resolved on SDS-PAGE and blotted with the anti-Myc antibody that detected Myc-Securin. The slow-migrating species represented the poly-ubiquitinated forms of Securin. For the reversine sample, the inhibitor was added to the kinase reaction containing Mps1 and Bub1$^{\Delta K}$–Bub3 prior to ATP addition.

The following figure supplement is available for figure 4:

**Figure supplement 1.** Characterization of monomeric and dimeric Mad2 in the APC/C$^{Cdc20}$ inhibition assay.

Cdc20$^{\Delta N60}$. This was the reason for the use of Cdc20$^{\Delta N60}$ in the APC/C assays described in the previous section, as Cdc20$^{\Delta N60}$ allowed us to eliminate the complications from this Bub1-independent function of Mps1.

It has been reported previously that overexpression of scMps1 in the budding yeast causes constitutive activation of the spindle checkpoint and hyperphosphorylation of scMad1 (*Hardwick et al., 1996*), although the functional relevance of this Mps1-dependent Mad1 phosphorylation has not been further explored. Incubation of the human Mad1$^E$–Mad2 complex with human Mps1 and ATP slightly reduced the gel mobility of Mad1$^E$ (*Figure 5C*), suggestive of Mad1 phosphorylation by Mps1. Mass spectrometry analysis identified 15 phosphorylation sites in Mad1$^E$, with 13 sites located in the flexible region N-terminal to the CTD (*Figure 5—figure supplement 1A*). Only two sites, T644 and T716, were located in Mad1 CTD, a domain known to be important for the spindle checkpoint (*Figure 5D*) (*Heinrich et al., 2014*; *Kruse et al., 2014*). Consistent with the initial discovery of human Mps1 as a dual-specificity kinase (*Mills et al., 1992*), a tyrosine residue in Mad1 was phosphorylated by Mps1.

We used the Mis12–Mad1 fusion protein to test the potential relevance of these phosphorylation sites in human cells (*Figure 5E* and *Figure 5—figure supplement 1B*). Mutating all identified phosphorylation sites in Mis12–Mad1 (YF-14A) abolished the mitotic arrest exerted by this fusion protein. Strikingly, mutating the two sites in the CTD individually (T644A and T716A) abrogated the function of Mis12–Mad1 more strongly than simultaneously mutating the 13 sites outside of the CTD. Mass spectrometry analysis of the endogenous Mad1 immunoprecipitated from mitotic HeLa cells did not produce coverage of the T644 or T716 regions. We thus generated a phospho-specific antibody against Mad1 phospho-T716, and confirmed that ectopically expressed Mad1 (with its MIM mutated to alanine prevent checkpoint inactivation; MIM/5A) was indeed phosphorylated at T716 during mitosis in human cells (*Figure 5F* and *Figure 5—figure supplement 1C*). We could not detect T716 phosphorylation on endogenous Mad1, presumably due to the low sensitivity of the antibody. Our attempt to generate phospho-specific antibody against Mad1 phospho-T644 failed, preventing us from verifying that Mad1 T644 could indeed be phosphorylated in human cells. Taken together, our results suggest that phosphorylation of Mad1 by Mps1 might be required for Mad1 function in human cells, with the two sites in the CTD being the most critical ones.

We tested whether phosphorylation of these Mad1 sites was involved in MCC assembly and APC/C inhibition. Mad1$^E$–Mad2 WT, but not the phosphorylation-deficient YF-14A mutant, supported Mps1-dependent MCC formation and APC/C inhibition (*Figure 5—figure supplement 1D*). Strikingly, mutation of a single phosphorylation site T716 in Mad1 abolished the stimulatory effect of Mps1. Thus, phosphorylation of Mad1 T716 by Mps1 enhances the formation of APC/C-inhibitory MCC in vitro. In contrast, the Mad1 T644A mutation did not affect the ability of Mps1 to stimulate MCC formation, suggesting that phosphorylation of this residue contributes to Mad1 function in vivo through other mechanisms. Surprisingly, mutating the RLK motif of Mad1 also abolished the phospho-Mad1-dependent APC/C inhibition, even in the absence of Bub1 (*Figure 5G*). This result indicates that the RLK motif has an additional function aside from binding to Bub1. There are two copies of the RLK motif in each Mad1 CTD homodimer. It is possible that one RLK copy binds to Bub1 while the other interacts with MCC components to facilitate MCC formation. Alternatively, the RLK mutation might have unintended structural consequences.

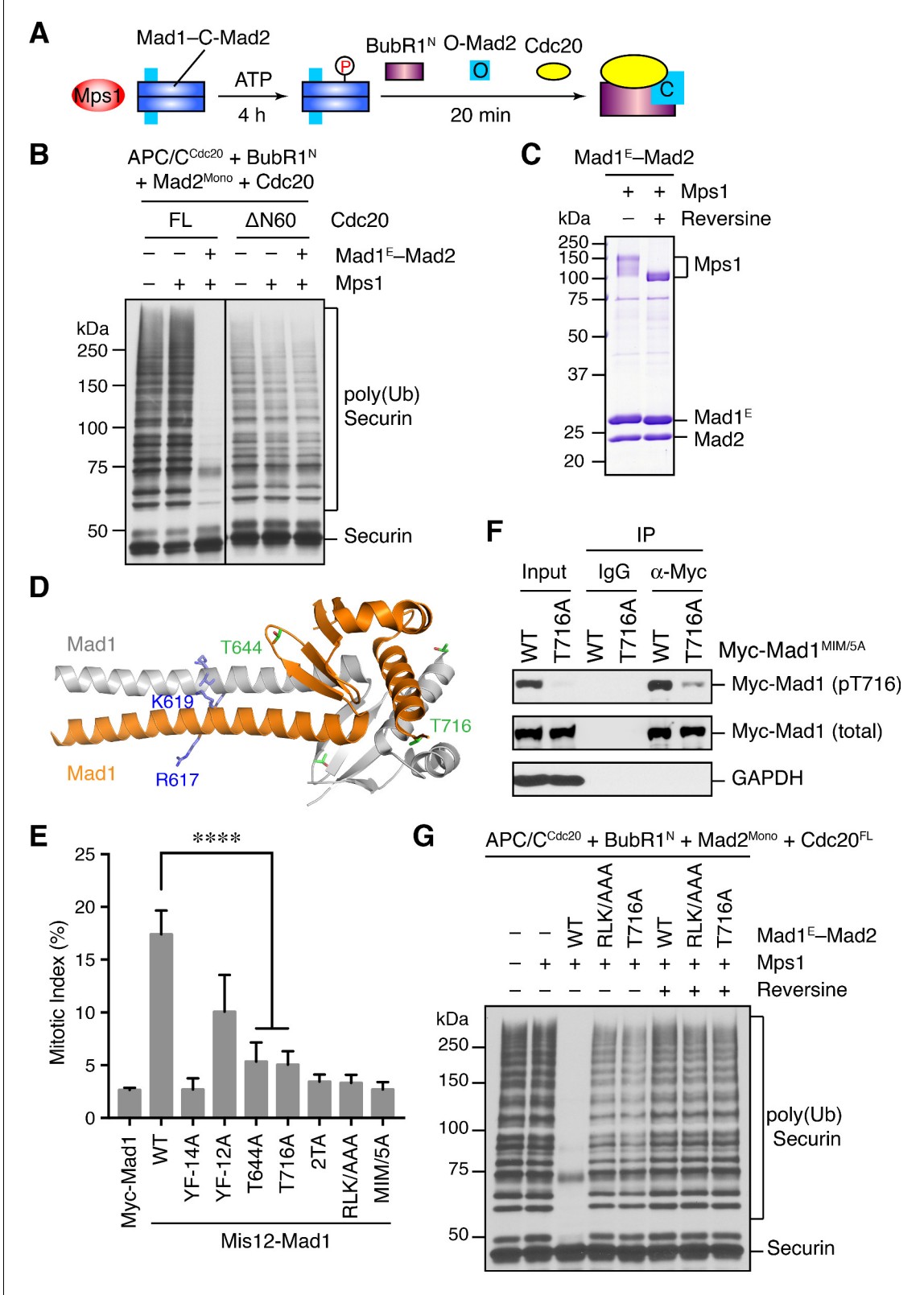

**Figure 5.** Phosphorylation of Mad1 T716 by Mps1 promotes MCC assembly and checkpoint signaling. (**A**) Schematic drawing of the assay examining MCC assembly facilitated by Mps1-phosphorylated Mad1–C-Mad2. (**B**) MCC assembly and APC/C inhibition were performed as depicted in (A). The ubiquitination reaction mixtures were resolved on SDS-PAGE and blotted with the anti-Myc antibody that detected Myc-Securin. The slow-migrating species represented the poly-ubiquitinated forms of Securin. FL, full-length. Cdc20$^{\Delta N60}$ lacks residues 1–60. (**C**) Recombinant human Mad1$^E$–Mad2

*Figure 5 continued on next page*

*Figure 5 continued*

complex was incubated with Mps1 in the presence or absence of reversine. The reactions were resolved on SDS-PAGE and stained with Coomassie. Note that Mps1 underwent autophosphorylation in the absence of reversine. (D) Cartoon drawing of the crystal structure of human Mad1$^{CTD}$ (PDB ID: 4DZO) with the RLK motif, T644, and T716 shown in sticks. (E) Mitotic indices of HeLa cells expressing the indicated Mis12–Mad1 fusion proteins. Error bars, s.d. (n = 5 independent experiments). ****p<0.0001; Student's t-test. (F) HeLa cells expressing Myc-Mad1 MIM/5A or MIM/5A;T716A were synchronized with thymidine and released into nocodazole-containing medium. MIM/5A, mutant with the Mad2 Interacting Motif replaced by alanine. Because overexpression of Mad1 inactivates the spindle checkpoint, we used Mad1 MIM/5A to prevent cells from escaping nocodazole-mediated mitotic arrest. Myc-Mad1 proteins were immunoprecipitated and blotted with indicated antibodies. (G) The MCC mixtures were prepared as depicted in (A) with the indicated Mad1$^E$ proteins, and then applied to the APC/C-dependent ubiquitination assay. The ubiquitination reaction mixtures were resolved on SDS-PAGE and blotted with the anti-Myc antibody that detected Myc-Securin. The slow-migrating species represented the poly-ubiquitinated forms of Securin. In reactions containing reversine, reversine was added prior to the addition of ATP.

The following figure supplement is available for figure 5:

**Figure supplement 1.** Characterization of Mad1 phosphorylation and its function in APC/C inhibition by MCC components.

## Mps1-mediated phosphorylation of Mad1 at T716 promotes Cdc20 binding

As shown in *Figure 5B*, Mps1 and Mad1$^E$–Mad2 only stimulated the APC/C$^{Cdc20}$-inhibitory activity of the MCC mixture containing Cdc20$^{FL}$, but not that of the mixture containing Cdc20$^{ΔN60}$. This suggests that Mps1-phosphorylated Mad1 might bind to the N-terminal tail of Cdc20, which contains two conserved basic motifs (BM1 and BM2) (*Figure 6A*). Indeed, the Mad1$^E$–Mad2 complex pre-treated with the kinase domain of Mps1 was efficiently pulled down by the His$_6$-tagged N-terminal fragment of Cdc20 (residues 1–170, Cdc20$^N$) (*Figure 6B*). Inhibition of Mps1 by reversine greatly reduced this interaction. Mutating T716, but not all other phosphorylation sites, greatly diminished this interaction. These results indicate that phosphorylation of Mad1 at T716 by Mps1 is necessary and sufficient for Cdc20$^N$ binding. Further mapping revealed a requirement for residues 26–37 of Cdc20 in phospho-Mad1 binding (*Figure 6C and D*), suggesting that BM1 ($^{27}$RWQRK$^{31}$) might be the binding site for Mad1 pT716. Because I-Mad2 binds to the Mad1-bound C-Mad2, this phosphorylation-dependent binding between Mad1 CTD and Cdc20$^N$ might place the MIM of Cdc20 in the vicinity of I-Mad2, promoting the formation of Cdc20–C-Mad2 complex, which can further bind BubR1 to form MCC.

## Discussion

Mps1 is a master kinase for spindle checkpoint activation. How Mps1 regulates checkpoint signaling is not fully understood. It is well established that Mps1 can phosphorylate Knl1 at multiple MELT motifs to recruit the Bub1–Bub3 complex to kinetochores (*Krenn et al., 2014*; *London et al., 2012*; *Primorac et al., 2013*; *Shepperd et al., 2012*; *Vleugel et al., 2013*; *Yamagishi et al., 2012*; *Zhang et al., 2014*). Mps1-dependent Bub1–Mad1 interaction has also been characterized in the budding and fission yeast (*London and Biggins, 2014*; *Mora-Santos et al., 2016*), but has so far not been reported in humans. In the current study, we have shown that after priming phosphorylation by Cdk1, human Mps1 can phosphorylate the conserved motif of human Bub1, which enables the binding of human Mad1. Therefore, Bub1 is another evolutionarily conserved substrate of Mps1. We have reconstituted phospho-Bub1–Mad1-dependent APC/C inhibition by MCC components in vitro and demonstrated a requirement for this interaction in the spindle checkpoint in human cells. In addition, we have identified novel phosphorylation events of Mad1 by Mps1, which are also required for checkpoint activation and APC/C inhibition. As a master kinase in the spindle checkpoint, Mps1 establishes a phosphorylation cascade by sequentially phosphorylating Knl1, Bub1, and Mad1 (*Figure 7A*). Such a multiple-target phosphorylation cascade reinforces the requirement for Mps1 during checkpoint activation, making checkpoint signaling completely dependent on and highly responsive to the kinase activity of Mps1.

The phosphorylated region of Bub1 and the RLK motif in Mad1 involved in phospho-peptide binding are both conserved from yeast to man, indicating a conserved binding mode. The checkpoint-dependent Bub1–Mad1 interaction is readily detectable in the budding yeast (*Brady and*

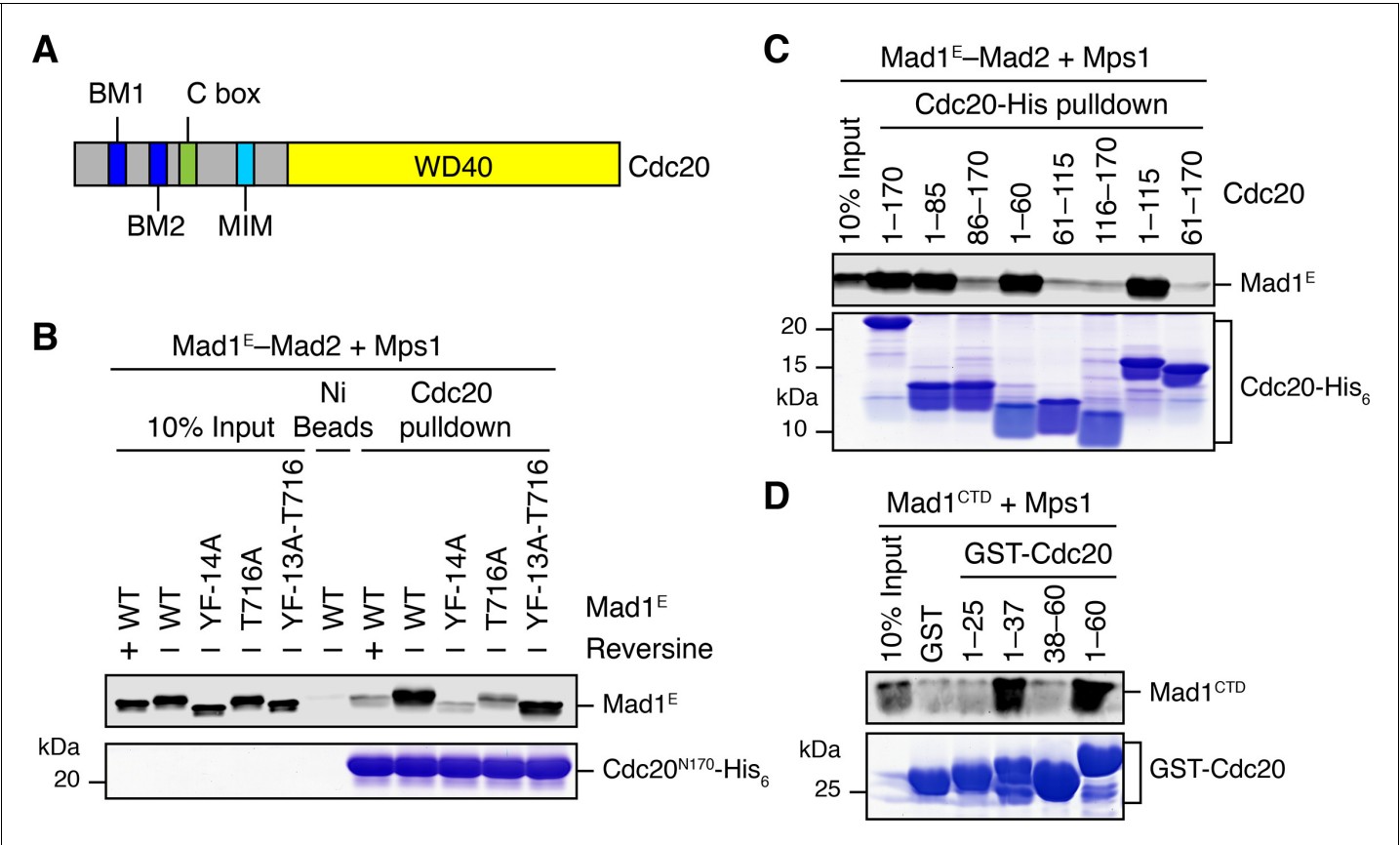

**Figure 6.** Phosphorylation of Mad1 T716 promotes its binding to Cdc20. (**A**) Domains and motifs of Cdc20. C box, a conserved APC/C-binding motif; MIM, Mad2-interacting motif; BM1, basic motif 1 ($^{27}$RWQRK$^{31}$); BM2, basic motif 2 ($^{54}$RTPGRTPGK$^{62}$). (**B**) In vitro pull-down of the indicated Mad1$^E$–Mad2 complexes (which had been pre-treated with the kinase domain of Mps1) by Ni$^{2+}$ beads bound to Cdc20$^{N170}$-His$_6$. The bait protein was stained with Coomassie, and the prey proteins bound to beads were blotted with the anti-Mad1 antibody. (**C**) In vitro pull-down of the Mad1$^E$–Mad2 complex (which had been pre-treated with the kinase domain of Mps1) by Ni$^{2+}$ beads bound to the indicated Cdc20-His$_6$ proteins. The bait proteins were stained with Coomassie, and the prey proteins bound to beads were blotted with the anti-Mad1 antibody. (**D**) In vitro pull-down of Mad1$^{CTD}$ (which had been pre-treated with the kinase domain of Mps1) by beads bound to the indicated GST-Cdc20 fragments. The bait proteins were stained with Coomassie, and the prey proteins bound to beads were blotted with the anti-Mad1 antibody.

*Hardwick, 2000*; *London and Biggins, 2014*). On the other hand, we have failed to detect the Bub1–Mad1 interaction in human cell lysates, even though we have demonstrated a role for Bub1 and the RLK motif of Mad1 in the kinetochore targeting of Mad1 (*Kim et al., 2012*). The yeast Bub1–Mad1 interaction requires only a single Mps1-dependent phosphorylation event at scBub1 T455, whereas the human Bub1–Mad1 interaction needs sequential phosphorylation at both S459 and T461 of human Bub1 by Cdk1 and Mps1 in vitro. In human cells, the Bub1–Mad1 interaction is likely buttressed by other factors at mitotic kinetochores. For example, the metazoan-specific Rod-Zwilch-ZW10 (RZZ) complex is particularly important for the Mad1 kinetochore localization (*Caldas et al., 2015*; *Silió et al., 2015*). It is possible that RZZ contributes to the Bub1–Mad1 interaction when all are anchored at kinetochores. The intrinsically weak affinity between Bub1 and Mad1, along with the difficulty of preserving the doubly phosphorylated Bub1 and other auxiliary interactions only present at intact kinetochores, may underlie the difficulty in detecting the Bub1–Mad1 interaction in human cell lysates. The Bub1–Mad1 interaction is also observed in C. elegans, although in that case the interaction involves different domains of Bub1 and Mad1 (*Moyle et al., 2014*). Thus, there are species-specific differences in this functionally important interaction.

Mps1 prefers to phosphorylate serine and threonine residues with a negatively charged residue at the −2 position (*Dou et al., 2011*). Similar to the MELT motifs of Knl1, the T644 site of Mad1

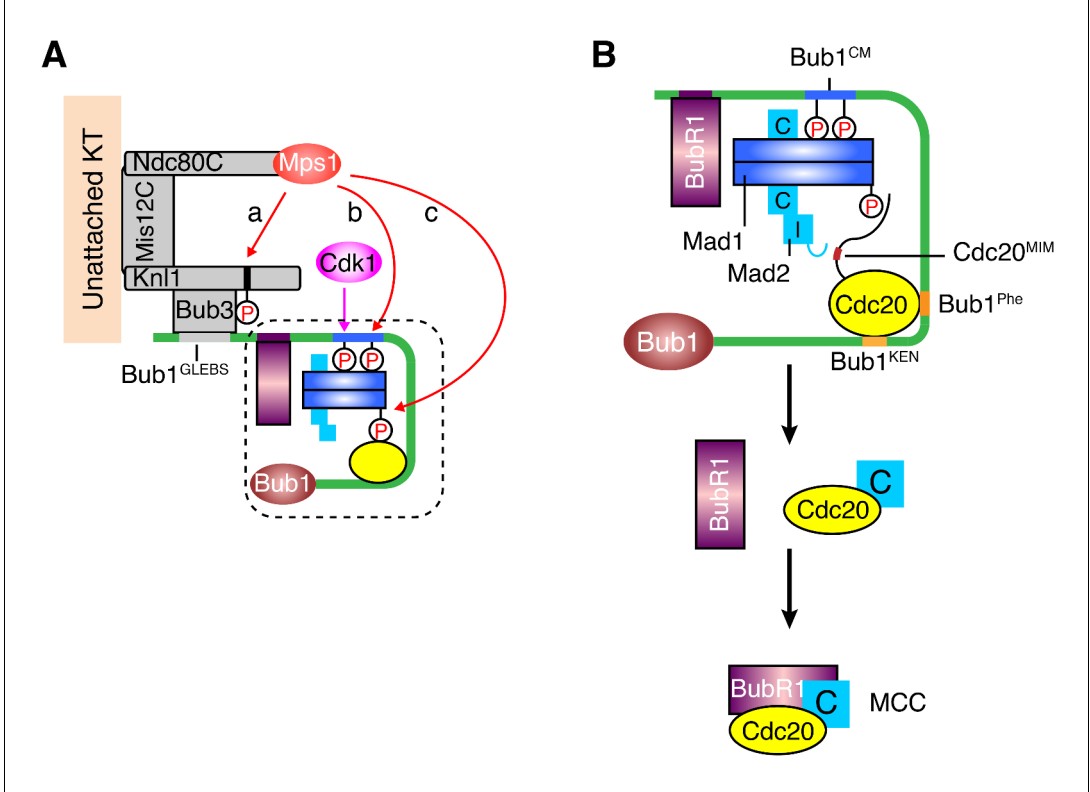

**Figure 7.** A sequential multi-target phosphorylation cascade by Mps1 promotes the assembly and activation of the Bub1–Mad1 scaffold. (A) Mps1 recognizes unattached kinetochores (KT) through its direct binding to the Ndc80 complex (Ndc80C). At kinetochores, Mps1 first phosphorylates Knl1 at multiple MELT motifs to recruit the Bub1–Bub3 complex (a). After Cdk1 phosphorylates Bub1 S459, Mps1 then phosphorylates Bub1 T461 (b). The doubly phosphorylated Bub1 conserved motif (CM) binds to and recruits the Mad1–C-Mad2 core complex. Mps1 then phosphorylates Mad1 at T716, and this phosphorylation enables Mad1 binding to Cdc20 (c). (B) The boxed region in (A) is magnified and shown with more molecular details here. The Mad1–C-Mad2 core complex bound to phosphorylated Bub1 CM can further recruit O-Mad2 and convert it to I-Mad2. The WD40 domain of Cdc20 is anchored to the Phe and KEN boxes of Bub1, whereas the N-terminal basic tail of Cdc20 is bound by the phosphorylated Mad1 CTD. This bipartite Cdc20-binding mode positions the MIM of Cdc20 close to I-Mad2, promoting the formation of the C-Mad2–Cdc20 complex. This binary complex can further bind to BubR1 (bound to Bub1 or from cytosol) to form MCC.

($^{641}$IDIT$^{644}$) identified in this study matches the substrate consensus of Mps1. Furthermore, phosphorylation of T461 in human Bub1 by Mps1 is enhanced by prior phosphorylation at S459 by Cdk1. The negatively charged phospho-S459 can serve the same purpose as the −2 acidic residue in the substrate recognition by Mps1. This type of Cdk1-primed Mps1 phosphorylation consensus likely exists in other Mps1 substrates. To our surprise, Mad1 T716 and adjacent residues do not match the consensus of Mps1 substrates. The MELT motifs of Knl1, Bub1 T461, and Mad1 T644 are all located in loops or flexible regions. In contrast, although it is surface exposed, Mad1 T716 resides in a well-folded α helix. We speculate that Mps1 might recognize Mad1 T716 through tertiary contacts, instead of a linear peptide motif. On the other hand, we cannot completely rule out the possibility that T716 is phosphorylated by a kinase other than Mps1 in vivo.

We do not know how phosphorylation of Mad1 T644 by Mps1 contributes to the spindle checkpoint, as it is not required for the reconstituted APC/C inhibition by MCC components in vitro. This phosphorylation does not appear to enhance the binding of phospho-Bub1 CM to Mad1. One possibility is that this phosphorylation is required for Mad1 CTD to interact with other regions of Bub1.

A key downstream activity of the spindle checkpoint is to inhibit APC/C$^{Cdc20}$. We have reconstituted Bub1–Mad1- and Cdk1/Mps1-dependent checkpoint inhibition of APC/C$^{Cdc20}$. Although we did not directly monitor the kinetics of MCC formation, it is very likely that the effects of Bub1–Mad1 and Mps1 on APC/C$^{Cdc20}$ inhibition are through promoting the assembly of MCC. We note that Bub1 has been reported to directly interact with BubR1 (*Overlack et al., 2015*). The Bub1–

BubR1 interaction has been reported to be dispensable for the spindle checkpoint, however (*Zhang et al., 2015*). BubR1[N] used in our assay does not contain the Bub1-binding region. Thus, our assays do not test whether the Bub1–BubR1 interaction can contribute to MCC assembly and APC/C inhibition.

How the upstream checkpoint proteins, after being recruited to kinetochores, direct the assembly of MCC is an unresolved question. Our in vitro reconstitution experiments have provided important insight into this question. Our results support a model in which Bub1–Mad1 acts as a scaffold to recruit and activate MCC components, thus promoting MCC assembly (*Figure 7B*). Two phosphorylation-dependent molecular interactions, the Bub1–Mad1 interaction and the Mad1–Cdc20 interaction, are critical for the assembly and proper function of the scaffold. We have dissected the individual contributions of these two interactions to MCC assembly and APC/C inhibition in vitro. For example, to define the function of Bub1–Mad1 in MCC assembly, we eliminated the effect of Mad1 phosphorylation by Mps1 through inhibiting Mps1 prior to the addition of Mad1, and through the use of Cdc20$^{\Delta N60}$, which no longer interacts with phosphorylated Mad1. Likewise, to assay the contribution of Mad1–Cdc20 to MCC assembly, we simply omitted Bub1 to eliminate its contribution. Although either interaction suffices to promote MCC assembly in vitro, disruption of either Bub1–Mad1 or Mad1–Cdc20 interactions in human cells largely abolished checkpoint activation Therefore, these two interactions likely cooperate with each other to promote checkpoint signaling in vivo. The Cdc20[N]-binding function of phosphorylated Mad1 CTD provides a possible explanation for the recent findings that Mad1 has additional checkpoint functions aside from recruiting and activating Mad2 (*Heinrich et al., 2014*; *Kruse et al., 2014*).

We propose that Bub1–Mad1 can simultaneously recruit I-Mad2, Cdc20, and BubR1 into the same large complex (*Figure 7B*). Within this complex, the WD40 domain of Cdc20 binds to the Phe and KEN boxes of Bub1, and the N-terminal tail of Cdc20 is anchored to phosphorylated Mad1 CTD. This bipartite anchorage may optimally position the MIM of Cdc20 in the vicinity of I-Mad2, allowing I-Mad2 to entrap the MIM. The resulting C-Mad2–Cdc20 complex can then bind to BubR1 to form MCC. Even though the Bub1–BubR1 interaction may not be critical for the checkpoint, the Bub1-bound BubR1 is in a good position to encounter Mad2–Cdc20 and to be incorporated into MCC.

In conclusion, combining in vitro reconstitution and cellular experiments, we have established the Bub1–Mad1 complex as an evolutionarily conserved catalytic platform for stimulating APC/C inhibition by MCC components. We have further demonstrated the importance of an Mps1-orchestrated phosphorylation cascade in forming and activating this catalytic platform. This multi-target, sequential phosphorylation cascade renders the checkpoint highly sensitive to changes in the kinase activity of Mps1. Because the kinetochore targeting of Mps1 itself is directly controlled by kinetochore-microtubule attachment (*Hiruma et al., 2015*; *Ji et al., 2015*), our findings further highlight a unique signal-transducing principle of the spindle checkpoint that not only enables signal amplification but also keeps the final signaling output responsive to kinetochore attachment status.

## Materials and methods

### Cell culture and transfection

HeLa Tet-On cells were initially purchased from Clontech (Mountain View, CA), and were cultured in Dulbecco's Modified Eagle Medium (DMEM, Invitrogen, Carlsbad, CA) supplemented with 10% fetal bovine serum and 2 mM L-glutamine. The cells were authenticated by genome sequencing and metaphase chromosome spread, and were routinely checked by 4',6-diamidino-2-phenylindole (DAPI) staining to ascertain that they were free of mycoplasma contamination. For G1/S synchronization, cells were treated with 2 mM thymidine (Sigma, St. Louis, MO) for 14–16 hr. For mitotic arrest, cells were treated with 330 nM or 1 μM nocodazole (Sigma) or 100 nM taxol (Sigma) for 12–14 hr. Other chemicals used in this study were: RO3306 (Sigma) at 20 μM, BI 2536 (Selleck Chemicals, Houston, TX) at 1 μM, MG132 (Selleck Chemicals) at 10 μM, okadaic acid (LC lab, Woburn, MA) at 500 nM, and reversine (Cayman Chemical, Ann Arbor, MI) at 1 μM.

For RNAi experiments, cells were transfected with siRNA oligonucleotides (Dharmacon, Lafayette, CO) using Lipofectamine RNAiMAX (Invitrogen) and harvested at 24–72 hr after transfection. The sequences of the siRNAs were (from 5' to 3'): siBub1-c, CCCAUUUGCCAGC

UCAAGC (*Jia et al., 2016*); siBub1-d, GAGUGAUCACGAUUUCUAA (*Jia et al., 2016*); siBubR1, GGACACAUUUAGAUGCACU (*Choi et al., 2016*); and siMad2, GCUUGUAACUACUGAUCUU (*Choi et al., 2016*).

Internal deletion mutants were constructed with overlapping extension polymerase chain reaction (PCR). Point mutations were introduced with QuikChange Lightening Site-Directed Mutagenesis kit (Agilent Technologies, Santa Clara, CA). All plasmids were verified by sequencing. Plasmid transfection was performed with the Effectene reagent (Qiagen, German Town, MD) according to the manufacturer's instructions.

For the generation of stable cell lines, HeLa Tet-On cells were transfected with pTRE2 vectors encoding siRNA-resistant wild-type or mutant Myc-Bub1$^{\Delta K}$ or GFP-Bub1 transgenes. Cells were treated with 200–300 µg/ml hygromycin (Clontech). Surviving clones were screened for Bub1 expression in the presence of 1 µg/ml doxycycline (Sigma). Clones that expressed Bub1$^{\Delta K}$ at a level similar to that of endogenous Bub1 were chosen for further analysis. Stable cell line clones were maintained with 100 µg/ml hygromycin.

## Antibodies

The following antibodies were used for immunoblotting and immunoprecipitation: anti-Tubulin (Sigma, T9026), anti-BubR1 [BD Biosciences; Clone 9/BUBR1 (RUO)], anti-Bub3 [mouse, BD Biosciences; Clone 31/Bub3 (RUO)], anti-Bub3 (rabbit, Sigma, B7811), anti-GAPDH (Cell Signaling, 14C10), anti-GFP (Roche), anti-Myc (Roche), and anti-Cdc20 (goat, Santa Cruz Biotechnology). Fluorescent secondary antibodies were purchased from Life Technologies. Antibodies against Mad2, APC3, Mad1, Cdc20 (pS153), and Bub1 were previously described (*Fang et al., 1998*; *Kim et al., 2012*; *Lin et al., 2014*; *Tang et al., 2001*). The Bub1 pS459 and pSpT antibodies and the Mad1 pT716 antibody were generated at an on-campus facility. Two Bub1 phospho-peptides, with the sequences of CKVQP[pS]PTVH and CKVQP[pS]P[pT]VHTK, and a Mad1 phospho-peptide, with the sequence of CELFSRQ[pT]VA, were used to immunize rabbits. The antibodies were affinity-purified with SulfoLink resins (Thermo Scientific) coupled to the phospho-peptides.

## Flow cytometry

Cells were synchronized at G1/S with thymidine and released into fresh medium containing 330 nM nocodazole for 13 hr before being harvested. For Plk1 inhibition, BI 2536 was added 1 hr prior to harvest. Collected cells were washed with PBS and fixed with pre-chilled 70% ethanol at −20°C overnight. After being washed with PBS, the fixed cells were permeabilized with 0.2% Triton X-100 in PBS for 20 min. Cells were then incubated with the anti-MPM2 antibody (1:100, Millipore) in PBS containing 1% BSA for 2 hr. After being washed with PBS, cells were incubated with fluorescent secondary antibodies (Invitrogen) in PBS containing 1% BSA for 30 min. After being washed with PBS, cells were stained with 20 µg/ml propidium iodide (Sigma) in PBS containing 200 µg/ml RNase (Qiagen). All staining steps were performed at room temperature. The samples were analyzed on a FACSCalibur flow cytometer (BD Biosciences). Data were processed with the FlowJo software. The mitotic index was defined as the percentage of cells that had 4N DNA content and were also positive for MPM2 staining. The graphs and statistic analysis were generated with the Prism software (GraphPad).

## Live-cell imaging

Cells stably expressing Bub1 transgenes were passaged onto four-well chambered coverglass (Lab-Tek) and transfected with the Bub1 siRNA. For visualization of chromosomes, Hoechst 33342 (50 ng/ml; Invitrogen) was added at 1 hr before imaging. In the case of taxol treatment, cells were treated with thymidine for 16 hr, and then released into fresh medium for 6 hr before taxol addition. Imaging was performed at 3 hr after taxol addition. Live-cell imaging was performed with a DeltaVision microscope (Applied Precision) equipped with an environmental chamber (37°C and 5% CO$_2$) and a CoolSNAP HQ2 camera (Roper Scientific). Differential interference contrast and fluorescent images were taken with a 40X objective (Olympus) at 10 min intervals for taxol-treated cells and at 5 min intervals for asynchronous cells. For taxol-treated cells, the mitotic duration was defined as the time from nuclear envelope breakdown (NEBD) to mitotic cell death or slippage. For asynchronous cells,

the mitotic duration was defined as the time from NEBD to anaphase onset. Images were processed with ImageJ, and the graphs were generated with the Prism software (GraphPad).

## Immunoprecipitation

Mitotic cells were harvested by shake-off and washed with PBS once. Cell pellets were lysed with the wash buffer (50 mM Tris-HCl, pH 8.0, 150 mM KCl, 0.1% NP-40, 2 mM $MgCl_2$, 5 mM NaF, 10 mM $\beta$-glycerophosphate, 1 mM DTT) supplemented with complete EDTA-free protease inhibitor cocktail (Roche), 500 nM okadaic acid, and 10 units/ml TurboNuclease (Accelagen). Cell lysates were cleared by centrifugation. Supernatants were incubated with antibody-coupled Protein A beads (Bio-Rad) at 4°C for 2 hr. Proteins bound to beads were released with the SDS loading buffer, and analyzed by Western blotting. For quantitative Western blots, immunoglobulin G (IgG) (H+L) conjugated with Dylight 680 or Dylight 800 (Cell Signaling) was used as the secondary antibody. The membrane strips were scanned with the Odyssey Infrared Imaging System (LI-COR, Lincoln, NE).

## Protein purification

Expression of GST-BubR1$^N$ (residues 1–370), GST-Mad2, GST-scBub1 fragments, GST-Mad1, and GST-scMad1 fragments in BL21(DE3) was induced by 0.1 mM isopropyl $\beta$-D-1-thiogalactopyranoside (IPTG) at 16°C overnight, after the $OD_{600}$ reached 0.8. Mad1 and scMad1 fragments (Mad1$^E$) containing the Mad2-interacting motif (MIM) were co-expressed with Mad2 and scMad2, respectively. Fragments of scBub1 were co-expressed with His$_6$-scMps1 (the kinase domain; residues 440–720). Harvested pellets were lysed in the wash buffer I [50 mM Tris-HCl, pH 8.0, 150 mM KCl, 0.1% (v/v) Triton X-100, 5% glycerol, 5 mM $\beta$-mercaptoethanol] supplemented with a protease inhibitor cocktail. After sonication, lysates were cleared by centrifugation at 4°C. Supernatants were filtered by 0.45 µm filter and incubated with pre-equilibrated Glutathione Sepharose 4B beads (GE Healthcare). Protein-bound beads were washed with 40 volumes of the wash buffer I. The proteins were then eluted with reduced glutathione (Sigma) or cleaved with the PreScission protease. Eluates were dialyzed to the QA buffer (25 mM Tris-HCl, pH 8.0, 20 mM NaCl, 3 mM DTT), loaded onto a Resource Q column (GE Healthcare), and then eluted with the QB buffer (25 mM Tris-HCl, pH 8.0, 1 M NaCl, 3 mM DTT). Peak fractions were pooled and further purified with a Superdex 200 size exclusion column (GE Healthcare). The relevant protein fractions were pooled, aliquoted, and snap-frozen for future experiments.

Expression of Cdc20-His$_6$ proteins containing the N-terminal regions of Cdc20 was induced by 1 mM isopropyl $\beta$-D-1-thiogalactopyranoside (IPTG) at 37 °C for 6 h. Harvested pellets were lysed in denaturing lysis buffer (100 mM $NaH_2PO_4$, 10 mM Tris-HCl, pH 8.0, 8 M Urea), and incubated overnight by gentle rocking at room temperature. After centrifugation, the supernatants were incubated with pre-equilibrated Ni$^{2+}$-NTA beads (Qiagen). Protein-bound beads were washed with 15 volumes of denaturing wash buffer (100 mM $NaH_2PO_4$, 10 mM Tris-HCl, pH 6.3, 8 M Urea). The proteins were then eluted with denaturing elution buffer (100 mM $NaH_2PO_4$, 10 mM Tris-HCl, pH 4.5, 8 M Urea). The eluted proteins were then dialyzed into the storage buffer (50 mM Tris, pH 8.0, 200 mM NaCl, 5% glycerol, 1 mM $\beta$-mercaptoethanol). The soluble proteins were snap-frozen for future experiments.

For expression of Bub1–Bub3, Mps1, Cdk1–Cyclin B, and Cdc20 in insect cells, baculoviruses encoding His$_6$-Strep-Bub1 (full-length or Bub1$^{\Delta K}$ containing residues 1–723), His$_6$-Bub3, His$_6$-Strep-Mps1 (full-length or the kinase domain containing residues 516–794), GST-Cdk1 (containing constitutively active mutations T14A/Y15F), His$_6$-Cyclin B1, or His$_6$-Cdc20 (full-length or the △N60 mutant lacking residues 1–60) were generated with the Bac-to-Bac system (Invitrogen). Bub1 and Cdk1 were co-expressed with Bub3 and Cyclin B1, respectively. Insect cells were infected with the appropriate viruses for 50 hr before being harvested. Cells expressing Bub1–Bub3, Mps1, or Cdk1–Cyclin B were lysed in the wash buffer I supplemented with a protease inhibitor cocktail. Cleared lysates were incubated with pre-equilibrated Strep-Tactin Superflow (Qiagen) or Glutathione Sepharose 4B beads. The protein-bound beads were washed and eluted with dethiobiotin (Sigma) or reduced glutathione. Eluates were dialyzed into the storage buffer I (50 mM Tris, pH 8.0, 150 mM KCl, 5% glycerol, 1 mM $\beta$-mercaptoethanol) and snap-frozen. Cells expressing His$_6$-Cdc20 were lysed in the wash buffer II (50 mM HEPES, pH 6.8, 250 mM KCl, 5% glycerol, 0.1% (v/v) Triton X-100, 5 mM $\beta$-mercaptoethanol and 10 mM imidazole) supplemented with a protease inhibitor cocktail and 10 units/ml

TurboNuclease. Cdc20 proteins were purified with $Ni^{2+}$-NTA beads (Qiagen) and eluted with 250 mM imidazole. The eluates were exchanged into the storage buffer II (40 mM HEPES, pH 6.8, 200 mM KCl, 5% glycerol and 2 mM DTT) with PD-10 columns (GE Healthcare).

Uba1, UbcH10, and Myc-Securin were purified as described previously (*Tang and Yu, 2004*). Ube2S protein was a gift from Dr. Michael Rape (University of California at Berkeley).

## In vitro kinase assays

For kinase assays, 1 μM Cdc20$^{\Delta N60}$, 40 μM Mad1$^{E}$–Mad2, or 400 nM Bub1$^{\Delta K}$–Bub3 was incubated with 40 nM Bub1, 500 nM Mps1, and/or 50 nM Cdk1–Cyclin B1 at room temperature for 30 min in the kinase buffer (25 mM Tris-HCl, pH 7.7, 100 mM NaCl, 10 mM MgCl$_2$, 5 mM NaF, 20 mM $\beta$-glycerophosphate, 0.1 mM Na$_3$VO$_4$ and 1 mM DTT) supplemented with 1 mM ATP. The reaction mixtures were stopped with the appropriate kinase inhibitors for further protein-binding assays, or quenched with the SDS loading buffer and analyzed by Coomassie blue staining or Western blotting.

## Protein-binding assays

Recombinant bait proteins or bacteria lysates containing bait proteins were incubated with the appropriate affinity beads. Bait-bound beads were then blocked with 2% fat-free milk when necessary before being mixed with 100 μl diluted prey proteins. After incubation, beads were washed and eluted with the SDS loading buffer. The bead-bound proteins were analyzed by Coomassie blue staining or Western blotting. For the quantification of band intensities, Coomassie-stained gels were scanned and quantified with the Odyssey Infrared Imaging System (LI-COR, Lincoln, NE). In cases that synthesized peptides were used as bait, peptides were immobilized on SulfoLink resin (Thermo Scientific, Waltham, MA) according to the manufacturer's instructions. The synthetic scBub1 peptides contained residues 449–473 of scBub1, various phospho-residues, and a cysteine at the N-terminus. All synthetic human Bub1 peptides, except the pS459 peptide, contained residue 455–477 of human Bub1, the appropriate phospho-residues, and a cysteine at the N-terminus. The Bub1 pS459 peptide contained residues 439–480, pT452/pS459, and a cysteine at the C-terminus.

All isothermal titration calorimetry (ITC) assays were performed with a MicroCal Omega ITC200 titration calorimeter (GE Life Sciences) at 20°C. Recombinant scMad1$^{CTD}$ protein or scMad1–scMad2 complexes were dialyzed into the HEPES Buffer I (25 mM HEPES, pH 7.5, 150 mM NaCl). Recombinant human Mad1$^{CTD}$ protein was dialyzed into the HEPES Buffer II (25 mM HEPES, pH 7.5, 50 mM NaCl). For each titration, 300 μl of scMad1 proteins (20 μM) or human Mad1 (50 μM) were added to the calorimeter cell. The scBub1 phospho-peptides or purified scMps1-phosphorylated scBub1$_{449-530}$ fragment (300 μM) in the HEPES Buffer I or the human Bub1 phospho-peptides (500 μM) in the HEPES Buffer II were injected with an injection syringe in 19 2-μl portions. The raw data were processed and fitted with the NITPIC software package (*Keller et al., 2012*).

## APC/C ubiquitination assays

APC/C was isolated from *Xenopus* egg extracts with the anti-APC3/Cdc27 antibody-coupled Protein A beads (Bio-Rad). For each reaction, 5 μl anti-APC3 beads were incubated with 60 μl extracts. The APC/C-bound beads were incubated with 1 μg recombinant human Cdc20. After washing away the unbound Cdc20, the APC/C$^{Cdc20}$ beads were incubated with the MCC mixture containing BubR1$^{N}$, Mad2, and Cdc20 with or without Bub1$^{\Delta K}$–Bub3 or Mad1$^{E}$–Mad2, and then washed before being added to ubiquitination reactions. The ubiquitination reaction mixture was prepared with the XB buffer (10 mM HEPES, pH 7.7, 100 mM KCl, 0.1 mM CaCl$_2$, 1 mM MgCl$_2$, and 50 mM sucrose) containing 150 μM bovine ubiquitin (Sigma), 5 μM Uba1, 750 nM UbcH10, 3 μM Ube2S, and 5 μM Myc-tagged human Securin, supplemented with 1X energy mixture (7.5 mM phosphocreatine, 1 mM ATP, 100 μM EGTA, and 1 mM MgCl$_2$). After incubation at room temperature with gentle shaking for 30 min, the reactions were quenched with the SDS loading buffer and analyzed by Western blotting.

To prepare the Bub1–Mad1 mixtures for reconstitution, 2.2 μM Bub1$^{\Delta K}$–Bub3 WT or SATA mutant was incubated with 75 nM Cdk1–Cyclin B1 and 900 nM Mps1 in the presence or absence of 10 μM reversine. After incubation at room temperature, the reactions were stopped by 20 μM RO3306 and 10μM reversine. (The two inhibitors were included in all buffers in the later steps.) The

inhibited kinase reactions were mixed with Mad1$^E$–Mad2 WT or RLK/AAA, resulting in the Bub1–Mad1 mixture (1.77 µM Bub1$^{\Delta K}$–Bub3 and 21 µM Mad1$^E$–Mad2). Then the Bub1–Mad1 mixture was diluted and supplemented with BubR1$^N$, Mad2$^{Mono}$, and Cdc20$^{\Delta N60}$, resulting in the MCC mixture (180 nM Bub1$^{\Delta K}$–Bub3, 2 µM Mad1$^E$–Mad2, 800 nM BubR1$^N$, 1.3 µM Mad2$^{Mono}$, and 675 nM Cdc20$^{\Delta N60}$). After a 20-min incubation, 8 µl MCC mixture (unless otherwise indicated) was diluted with 17 µl XB buffer and applied to the 5-µl APC/C$^{Cdc20}$ beads.

To prepare the Mad1–Mps1 mixture for Bub1-independent APC/C inhibition, 8 µM WT or mutant Mad1$^E$–Mad2 was incubated with 1 µM Mps1 in the presence or absence of reversine. After incubation at room temperature, the reactions were stopped by 10 µM reversine. (The inhibitor was included in all buffers in the later steps.) Then the Mad1–Mps1 mixtures were combined with BubR1$^N$, Mad2$^{Mono}$, and Cdc20$^{\Delta N60}$, resulting in the MCC mixture (4 µM Mad1$^E$–Mad2, 800 nM BubR1$^N$, 1 µM Mad2$^{Mono}$, and 600 nM Cdc20$^{\Delta N60}$). After a 20-min incubation, 8 µl MCC mixture was diluted with 17 µl buffer and applied to the 5-µl APC/C$^{Cdc20}$ beads.

## Acknowledgements

We thank Sue Biggins and Nitobe London for providing the scBub1, scMps1, scMad1, and scMad2 constructs and for helpful discussion, Michael Rape for providing recombinant Ube2S, Chad Brautigam for assistance with isothermal titration calorimetry, and Haydn Ball for peptide synthesis. We are grateful to the UT Southwestern Proteomics Core Facility for mass spectrometry analysis and the Animal Resource Center at UT Southwestern for assistance in generating the phospho-specific antibodies. This study is supported by grants from the Cancer Prevention and Research Institute of Texas (RP110465-P3 and RP120717-P2) and the Welch Foundation (I-1441). HY is an Investigator with the Howard Hughes Medical Institute.

## Additional information

### Funding

| Funder | Grant reference number | Author |
|---|---|---|
| Cancer Prevention and Research Institute of Texas | RP110465-P3 | Hongtao Yu |
| Cancer Prevention and Research Institute of Texas | RP120717-P2 | Hongtao Yu |
| Welch Foundation | I-1441 | Hongtao Yu |

The funders had no role in study design, data collection and interpretation, or the decision to submit the work for publication.

### Author contributions

ZJ, Conceptualization, Data curation, Formal analysis, Investigation, Methodology, Writing—original draft, Writing—review and editing; HG, Conceptualization, Data curation, Formal analysis, Investigation, Methodology, Writing—review and editing; LJ, BL, Conceptualization, Data curation, Formal analysis, Investigation, Methodology; HY, Conceptualization, Formal analysis, Supervision, Writing—original draft, Project administration, Writing—review and editing

### Author ORCIDs

Hongtao Yu, http://orcid.org/0000-0002-8861-049X

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
