## [Decision Letter]

Thank you for submitting your article "A sequential multi-target Mps1 phosphorylation cascade promotes spindle checkpoint signaling" for consideration by *eLife*. Your article has been favorably evaluated by Ivan Dikic (Senior Editor) and three reviewers, one of whom is a member of our Board of Reviewing Editors. The reviewers have opted to remain anonymous.

The reviewers have discussed the reviews with one another and the Reviewing Editor has drafted this decision to help you prepare a revised submission.

Summary:

In the past few years, the players of the mitotic checkpoint and their hierarchy of action have been well established. However, a mechanistic understanding of the catalytic assembly of the MCC at the kinetochore is still missing. In this study the authors provide important insights in the role of the kinases Cdk1 and Mps1 in promoting the interaction between Bub1 and Mad1 in human cells. In addition, they reveal a new function of MAD1 in recruiting Cdc20. Through in vitro reconstitution, they demonstrate that these interactions help to bring together the core components of the MCC, enabling its assembly. The work is of high quality and the findings are of great interest for the SAC field.

Essential revisions:

1) One concern is with the data in Figure 3, where we would be interested in seeing the same experiment without Plk1 inhibition – for better comparison to Figure 1.

2) It seems that experiments in Figure 1 may have been performed only once (low n). If so, please provide data for triplicates.

---

## [Author Response]

Essential revisions:

1) One concern is with the data in Figure 3, where we would be interested in seeing the same experiment without Plk1 inhibition – for better comparison to Figure 1.

The original experiments (three independent repeats) in fact included samples without Plk1 inhibition. These samples were not shown in the original submission to avoid repetition. As requested by the reviewers, we have now shown the entire dataset in Figure 3. This is a great suggestion. Including these data is indeed informative.

2) It seems that experiments in Figure 1 may have been performed only once (low n). If so, please provide data for triplicates.

The original experiment with mCherry-Bub1 stable cell lines was performed only once by a former graduate student Luying Jia in the lab. Because these cell lines did not behave properly after long-term storage in liquid nitrogen, we have performed new experiments in triplicates using stable cell lines expressing GFP-Bub1 proteins (Figure 1). The results were consistent with the original dataset. With the large sample size (n>150 for each sample), we could now see significant differences between the Bub1 FL and ΔK samples and between the ΔK WT and S459A samples.